# Green IoT Event Detection for Carbon-Emission Monitoring in Sensor Networks

**DOI:** 10.3390/s24010162

**Published:** 2023-12-27

**Authors:** Cormac D. Fay, Brian Corcoran, Dermot Diamond

**Affiliations:** 1SMART Infrastructure Facility, Engineering and Information Sciences, University of Wollongong, Wollongong, NSW 2522, Australia; 2School of Mechanical and Manufacturing Engineering, Faculty of Engineering and Computing, Dublin City University, Glasnevin, D09 V209 Dublin, Ireland; brian.corcoran@dcu.ie; 3Insight Centre for Data Analytics, Dublin City University, Glasnevin, D09 V209 Dublin, Ireland; dermot.diamond@dcu.ie

**Keywords:** chemical sensing, electricity metering, event detection, TinyML, IoT, green, carbon emissions

## Abstract

This research addresses the intersection of low-power microcontroller technology and binary classification of events in the context of carbon-emission reduction. The study introduces an innovative approach leveraging microcontrollers for real-time event detection in a homogeneous hardware/firmware manner and faced with limited resources. This showcases their efficiency in processing sensor data and reducing power consumption without the need for extensive training sets. Two case studies focusing on landfill CO${}_{2}$ emissions and home energy usage demonstrate the feasibility and effectiveness of this approach. The findings highlight significant power savings achieved by minimizing data transmission during non-event periods (94.8–99.8%), in addition to presenting a sustainable alternative to traditional resource-intensive AI/ML platforms that comparatively draw and produce 20,000 times the amount of power and carbon emissions, respectively.

## 1. Introduction

Climate change stands as one of the most pressing challenges confronting contemporary society. International bodies such as the United Nations Framework Convention on Climate Change (UNFCCC), the Intergovernmental Panel on Climate Change (IPCC), and national/regional Environmental Protection Agencies (EPAs) have meticulously identified the primary sources of carbon emissions. These emissions predominantly emanate from sectors such as electricity and heat, transport, manufacturing and construction, agriculture, and waste disposal [1,2,3]. The imperative to accurately monitor and quantify carbon emissions on a continuous basis for informing mitigation efforts has spurred the deployment of sensor networks. This focus has garnered considerable attention from governing bodies, industries, and academic institutions over an extended period [4,5,6,7].

In the domain of wireless sensor networks (WSNs), a noticeable trend is the integration of artificial intelligence (AI) and machine-learning (ML) models for processing and interpreting the steady influx of time-series data [8,9,10,11]. The development and application of these models often necessitate a significant volume of data points, ranging from thousands [12] to hundreds of thousands [13]. It is worth noting that non-eventful data holds almost equal importance to eventful data for the training and validation of these models. As more advanced approaches gain ground over traditional ML algorithms in WSNs such as deep learning [14,15], the demand for data generation on sensor nodes has surged, sometimes reaching into the millions [16]. However, the drawback of this approach, particularly concerning the tracking of carbon emissions, lies in the energy-intensive systems required to power standard and specialized server farms [17]. These systems can paradoxically exacerbate the very carbon emissions that sensor networks aim to alleviate. Studies have estimated that data centers alone contribute significantly to climate change, with emissions as high as 100 megatons of CO${}_{2}$ per year [18,19,20]. For instance, Strubell et al. [21] estimated the financial and environmental costs associated with training/tuning neural network models; they concluded that the carbon footprint increases proportionally with model size. Other works have focused on the environmental impacts of AI [22] and offer insights into calculating the carbon footprint of machine learning [23,24,25].

Although the application of ML systems for event classification in time-series data has yielded considerable benefits [26,27], the prevailing design philosophy for sensors has shifted towards configuring sensor nodes as basic devices primarily, if not entirely, responsible for data sampling and transmission. This approach offers several advantages, including efficient and standardized WSN design and the integration of advanced classification algorithms using growing libraries of big data analytics. However, a notable consequence of this approach is the increased demand for sensor node resources for data generation and transmission. Coupled with the need for higher sampling frequencies to train ML models for event detection, this approach significantly raises power consumption and, consequently, reduces the operational lifespan of these devices.

Concurrently, advances in computing hardware, such as edge computing, have spurred progress in microcontroller unit (MCU) technology [28,29]. These advancements have expanded the processing capabilities of microcontrollers while maintaining their energy-efficiency attributes [30]. This transformation can be partly attributed to the ongoing development of personal devices, such as wearables, smartphones, tablets, and more. Such devices have witnessed substantial enhancements in processing power and extended lifespans. Regrettably, these achievements have somewhat overshadowed the potential of microcontrollers in favor of the migration toward ML-based solutions. This shift raises a second paradox: prioritizing the computational demands of data transmission inadvertently shortens the operational lifespan of sensors.

The overarching challenges associated with the current/traditional server/desktop-based computers approach for machine learning have been well recognized, primarily in terms of their energy consumption, carbon footprint, and operational costs, which has consequently given rise to a young but growing paradigm shift (deemed TinyML) involving utilizing microcontrollers for data analysis as an alternative/solution [31,32,33,34,35]. Some examples where AI/ML can complement the efficiency of microcontrollers without compromising environmental sustainability include life prediction of turbofan engines [36], gas leakage detection [37], driver drowsiness detector [38], water leak detection [35], or fruit variety classification [39].

Microcontrollers play a crucial role in Internet of Things (IoT) applications [40]. Their compact size, low power consumption, and ability to integrate seamlessly with sensors make them fundamental components in IoT ecosystems [41,42,43,44]. With their embedded processing capabilities, microcontrollers enable real-time data acquisition and local decision-making, reducing the need for continuous communication with centralized servers [28,33,45,46,47]. This not only enhances system responsiveness but also contributes to energy efficiency, a critical factor in the design of sustainable IoT solutions [48,49]. Moreover, the cost-effectiveness and accessibility of microcontrollers have spurred innovation across industries, from smart homes and healthcare to industrial automation [50,51,52,53,54,55,56,57,58,59]. Understanding the pivotal role of microcontrollers in IoT is essential for harnessing their full potential in addressing contemporary challenges and advancing the capabilities of connected devices. In the context of carbon-emission monitoring, microcontrollers play a vital role in monitoring solutions. For instance, Brown et al. present a low-cost, microcontroller-based CO${}_{2}$ concentration data logger that can be used for field deployment [60]. Devan et al. [61] used a microcontroller to monitor carbon emissions in air pollutants. Afroz et al. used an ESP32 to measure CO${}_{2}$ emissions in urban areas [62]. Additionally, Vargas-Sansalvador [63] reported an interesting approach to CO${}_{2}$ sensing using an LED as the light detector with a colorimetric indicator. Although there are many examples in the literature [64], it is clear that microcontrollers have been used extensively for carbon-emission sensing, showcasing the importance of their use in monitoring systems.

An optimally designed WSN should, at least partially, offload low-level event detection tasks to sensor nodes. This approach capitalizes on the evolving processing capabilities of sensor technology, alleviating the necessity of expending energy reserves on data transmission and, therefore, server/cloud carbon emissions, specifically when no events occur. This is a more efficient utilization model of power resources, particularly for sensors expected to operate for extended periods of time, if not indefinitely, and for sensors that have already been deployed for several years. Additionally, it is crucial to recognize that the interpretation of data is often context-specific. Consequently, it raises a compelling question regarding what operations of event detection can be feasibly implemented on low-powered devices that are suitable for application across diverse domains.

Although the application of microcontroller-based data processing offers significant advantages, it also faces several challenges, including hardware heterogeneity, MCU architectures, resource constraints, limited memory management, and software interoperability between devices. When such an overarching challenge is presented, it is often valuable to examine fundamental elements and explore solutions capable of producing workable models in compliance with such constraints. In this study, we explore the potential for implementing a heterogeneous event detection approach based on binary classification, which can be adapted to different data sources within the realm of monitoring carbon emissions. Our investigation revolves around two distinct case studies closely tied to emission sectors. The first is a complex dataset of domestic energy monitoring, representing the electricity and heating sector. The second is less complex data comprised of environmental chemical sensing, focused on monitoring CO${}_{2}$ emissions within the waste disposal sector. Leveraging algorithms that can be efficiently implemented on standard microcontrollers, our objective is to detect specific events of interest in these crucial sectors to best demonstrate a heterogeneous approach across vastly differing data sets across two distinct domains from a sensory viewpoint.

## 2. Materials and Methods

### 2.1. Sensing Architecture

The sensing model employed during this study is illustrated in Figure 1, which illustrates our network setup following an Extended Start Topology arrangement—a spoke–hub distribution paradigm. It can be seen from the figure that although the applications differ significantly, a similar framework can be adopted for these distinctly different sensing applications. Although the actual sensors were completely different (IR absorbance and electrical induction for the environmental and home energy sensors, respectively), both sensing units were equipped with wireless communications capability to stream data to a local base station. For the environmental data, the device communicated via the Global System for Mobile (GSM) communications network. This was ideally suited for environmental systems as, in most cases, such devices are deployed in remote locations without access to local hot spots (e.g., WiFi or locally placed base stations). Furthermore, the infrastructure allowed for many sensing nodes (i.e., scope), and the data were sent via SMS, which required a quick energy burst for reduction of energy; see [65] for more details. For the home energy monitoring, the data were transmitted via Zigbee-based protocol to a local receiver (base station), and the household’s Internet connection was used for data delivery. The advantages of using Zigbee were based on its low power consumption, suited our data throughput, and allowed scope for several network topologies. In both cases, the data were stored on a high-end remote server and were readily available via the Internet, either directly to authorized users in RAW format or the form of an easy-to-use portal page (Figure 1). For visualization, simple color coding was used—if an event occurred, the data point was colored red; if not, it was green. This was helpful in visualizing event patterns to explore whether predictive models might be feasible.

### 2.2. Case Study I: Environmental Chemical Sensing

Our studies in the area of gas sensing have resulted in several in-house designed, developed, and deployed gas sensing systems capable of autonomously performing measurements and reporting data wirelessly to online repositories [65,66,67,68]. For this study, we considered a platform equipped with an infrared (IR)-based carbon dioxide (CO${}_{2}$) gas sensor (Dynament, Process Sensing Technologies, Cambridgeshire, UK, processsensing.com (accessed on 21 November 2023)), which had a dynamic sensing range of 0–20% v/v. The platform was calibrated by varying CO${}_{2}$/air proportions via mass flow controllers (Cole Parmer, Cambridgeshire, UK, coleparmer.com accessed on 21 November 2023), using a certified CO${}_{2}$ supply (Scott Specialty Gases, Breda, Netherlands) and ambient air via an air compressor (Werther International, Reggio Emilia, Italy). A reference instrument (GA 2000 Plus, Geotech, Keison, Chelmsford, UK) was used to validate the sample gas composition in-line for validation purposes. Using this approach, a calibration plot was achieved from which the system’s response to various CO${}_{2}$ samples could be obtained and unknowns estimated.

An ultra-low-power microcontroller (MSP430, F449, Texas Instruments, Austin, TX, USA) was employed for autonomous operational purposes. The platform was deployed for 12 months, during which time it reported CO${}_{2}$ emission levels from a perimeter borehole well on a waste landfill site to an online database every 6 h via GSM. The sampling frequency set by legislation is 4 measurements per year [69] In terms of capturing events, we increased this to 4 measurements per day to investigate whether events were being missed either daily or outside the governing monitoring frequency of once per quarter. This allows us to investigate the occurrence of events daily while extending the lifetime of the system for the 1-year deployment. A complete description of the platform and accompanying online services is available through [65]. Please note that for December 2010 and January 2011 (surrounding day 300), the low atmospheric temperature depleted the battery faster than expected, and coupled with site closure, this resulted in a loss of data.

### 2.3. Case Study II: Energy Monitoring

In parallel with the environmental deployment, a single-phase electricity monitor (ZEM-30, Episensor Ltd., Limerick, Ireland) was deployed/clamped at the mains input line of the household’s electricity supply and reported the integrated household energy usage for the same duration as for the environmental deployment. This was operated using a system-on-chip (SoC) microcontroller (Ember, EM250, Westlake Village, CA, USA). The principle behind this measurement was to monitor the current drawn by the household by means of an induction sensor. Data were transmitted via the sensor’s integrated Zigbee radio transceiver at a sampling period of 1.5 s to a local base station, from which it was uploaded to a central data repository every hour by means of a custom software application. Based on early examination of the data, e.g., the training set in Figure 4 (discussed later), it was found that the appliance signature of interest operated between 6 and 33 min. A sampling period of 1.5 min was set to safely allow for event detection and comply with the Shannon–Hartley theorem.

Using the differential approach, the times at which threshold crossing was detected (shower on), coupled with the corresponding negative crossing (shower off), were first compiled and recorded. After this, the RMS current during these periods was taken to represent shower events and statistics were similarly compiled.

### 2.4. Binary Classification Algorithm

The implemented C code, provided in Table A1 in Section A.1, outlines a robust algorithm for event detection using microcontrollers. The code defines key data structures such as DataPoint, GlobalStats, Event, and EventStats to organize and store relevant information. Functions like updateGlobalStats, updateEvent, and updateGlobalEventStats facilitate the dynamic updating of global and event-specific statistics as new data points are processed. The main program operates in a continuous loop on the main thread for real-time data acquisition. It retrieves data points, identifies events based on a specified binary classification, and updates statistics accordingly. The resulting metrics, including measurements, percentage above threshold, summation, maximum, minimum, and average values, as well as event-related statistics like the total number of events, average events per day, week, and month, maximum and minimum event values, and average event duration, are print to screen in this iteration (for clarity). This well-structured and efficient implementation serves as a foundation for autonomous event detection, showcasing the capabilities of microcontrollers in real-time monitoring scenarios.

## 3. Results

### 3.1. Case Study I: Environmental CO${}_{2}$ Emissions

#### 3.1.1. Calibration

Figure 2 presents the results of the calibration routine described previously. The left part of the figure shows the time-series captured by the system in response to the changing exposure to CO${}_{2}$ concentration gas. The sections in which the response was observed were extracted and appear in the figure’s inset. The time at which a steady state was observed was selected (1 to 2.5 min) to represent the response of the sensor. This informed the programming of the system with a 60-s warm-up period before sampling to ensure a steady-state response. The average and standard deviation of the steady-state response were calculated and plotted against reported CO${}_{2}$ concentrations by the reference instrument; see Figure 2 (Right). Here, the trend appeared to be linear, which was confirmed by an excellent linear model (R${}^{2}$ = 0.998) and allowed for future ADC readings to be converted to CO${}_{2}$ concentrations.

#### 3.1.2. Trial Data

The data collected during the 1-year deployment period for environmental CO${}_{2}$ emissions is presented in Figure 3. Upon initial visual inspection, one can readily observe a significant number of spikes in the dataset that exceed the legal emission threshold, indicated by the dashed red line. These spikes, often associated with increased CO${}_{2}$ concentrations, are considered events and are of particular interest to regulatory bodies, such as the EPA.

However, it is essential to acknowledge that during December and January (centered around day 300), the sensor system faced operational challenges due to the exceptionally low atmospheric temperatures. These harsh environmental conditions led to a faster depletion of the battery than initially anticipated, and in conjunction with the temporary site closure during this period, it resulted in the loss of some data.

Although it remains possible for an analyst to manually classify these events, particularly with a frequency of approximately four events per day, there is a clear advantage in automating this process. Automatic event classification not only enhances efficiency but also ensures that event identification is not reliant on the availability of human resources. The capabilities of the microcontroller utilized in this study offer a viable solution for such automatic analysis, providing an efficient means to detect and classify these events. This represents a practical application of microcontrollers in enhancing data processing capabilities within sensor networks, further reinforcing their potential utility in environmental monitoring.

### 3.2. Case Study II: Energy Monitoring

#### 3.2.1. Labelled Data Set

Figure 4 presents the training dataset of the home energy monitor during a typical day over a period of 12 h, i.e., from 6 a.m. to 6 p.m. in which all significant appliances were in use. Both the harvested RMS current datasets are present along with the first differential profile. It can be seen that as an appliance is powered on/off, coincidental positive and negative spikes in the differential data occur. For the shower, these spikes are particularly large, in contrast to the cumulative effect of several appliances being turned on simultaneously. For example, the relatively large RMS current recorded for the simultaneous use of the oven and kettle (far right, Figure 4 top) can be easily seen to relate to two appliances from the differential plot. Hence, the combined use of the RMS current and the differential plot can significantly reduce the occurrences of false positives and yield a much more accurate dataset for event detection.

For this study, the training set has focused on devices that have featured two criteria: (1) they draw a recognizable current for processing, and (2) they remain a static household device during the monitoring time period. The training set was therefore modeled on all constant devices, yet does not take into account devices more dynamic in nature, such as phone chargers, which have relatively insignificant power signatures and a much lower probability for misclassification of a single ‘heavy’ appliance. From Figure 4, it is clear that the most significant ‘event’ or largest power drawing appliance corresponded to the electric shower, then the electric kettle, and so forth. However, for simplicity, this study only considers the electric shower as a model for event detection, although the approach can be successfully employed to identify other appliances. Consequently, the detection of the most prominent power-demanding appliance (shower) is considered for our binary classification event detection. A threshold of 25 A is chosen on the differential dataset as it sufficiently classifies the shower use and is equally distant in value from other appliances, even when two power-demanding appliances (oven and kettle) are powered together.

#### 3.2.2. Trial Data

Figure 5 displays the complete 1-year data series for home energy consumption. A visual examination of this extensive dataset proves to be a challenging task due to the sheer density of data points. To glean meaningful insights, it becomes evident that a daily temporal resolution, as depicted in Figure 4, offers a more manageable approach. What stands out prominently are the numerous spikes in the dataset throughout the year, which are likely attributable to high-power consumption appliances.

The training data drawn from this dataset often prioritizes the identification of these prominent power-draining events before delving into the subtler nuances of energy consumption. Additionally, it is worth noting that the time surrounding days 300 to 315 corresponds to a period when the household was on holiday during December. This observation raises an intriguing possibility: the dataset could serve as a foundation for intrusion detection. Identifying unusual energy consumption patterns during times of expected inactivity could enhance security systems.

Moreover, it is worth highlighting the potential security implications associated with such detailed data. As recent incidents have shown, centrally stored data are susceptible to hacking and misuse (cybersecurity concerns). The ability to determine when a household is vacant based on energy consumption patterns could make it a target for nefarious activities, including home burglaries. This further underscores the importance of considering the security and privacy aspects when dealing with highly granular energy consumption data.

Although the 1-year energy trial data presents significant challenges in terms of data density and security, it also holds immense promise. Microcontrollers, with their low-power processing capabilities, provide an efficient means of extracting valuable insights from this dataset. Importantly, they may allow for focusing on the primary power-draining appliances, which aligns with the common user priority of identifying and managing the devices contributing most to their electricity bills and associated carbon emissions. This suggests that complex central databases and energy-intensive AI/ML systems may not always be necessary, particularly when addressing primary energy consumption concerns. Furthermore, this discussion reinforces the argument for data decentralization, especially when considering home user data and cybersecurity concerns. The distributed approach not only enhances user privacy and security but also offers a more streamlined and efficient means of managing energy consumption.

## 4. Discussion

### 4.1. Global Analysis

The initial sweep involved gathering global summary information on the total number of measurements, maximum/minimum/average sensed value, and percentage of measurements above critical thresholds. These global statistics are presented in Table 1. For the environmental CO${}_{2}$ emissions, the recorded concentration for the entire year was, on average, 1.41% v/v, which is surprisingly very close to the 1.5% v/v legal emission limit permitted by legislation [69]. This trigger level for CO${}_{2}$ emissions by the governing body laid the basis of our binary classification event detection. Furthermore, it seems that for ca. 28.5% of the total recorded measurements, the borehole well emission level was in excess of the regulatory limit.

For the home energy data, the data reflects the RMS current measurement via the reported data, i.e., not the differential dataset. The differential set was used to detect the bottom statistic in the table. Based on the training set in Figure 4, it is evident why two different thresholds were chosen for identifying shower use, i.e., due to the offset effect of other appliances. The offset also accounts for differences in the calculated maximum values. Curiously, the minimum was not 0 A, as one may expect; this is a good indication that the background appliances (e.g., refrigerators/freezers) are constantly in operation. The average RMS current of the household was 3.39 A over the entire year, and this can be used to estimate average costs and contributions to carbon footprint.

One point to note for both deployment data is the capability of the microcontroller to calculate the average value. First, both systems were capable of handling 4-byte floating point numbers as part of the microcontroller’s arithmetic logic units (ALUs). This meant that values from 1.2 × $10^{-38}$ to 3.4 × $10^{38}$ were possible, with calculations therein. This is clearly in range for calculation of the average and demonstrated through the summation field in the table. It is also worth noting that many microcontrollers are equipped with a carry flag, allowing for larger arithmetic ranges where required. The capability of generating such information by the microcontroller demonstrates its ability to provide meaningful information without the need for cloud-based servers and heavy computational processing.

One point to note is that fading problems affect data transmission. For the environmental data, Table 1 shows 1121 packets/measurements, yet an ideal packet count should be 1460 at 6 h/day over the year with a packet loss of 23.2%. This can be accounted for by battery depletion during the winter months, as discussed earlier. For the energy monitoring, the gateway was placed in the adjacent room to the monitor to ensure data reception. Ideally, at the set sampling frequency, 350,400 packets should be received over the year, yet 344,922 were accounted for. This yields a 98.4% successful transmission and a 1.6% packet loss, which could be due to attenuation by householder activity. Although the household sensor was equipped with CRC checking per transmitted packet coupled with 3 retries, the environmental sensor was not equipped with this capability due to the larger amount of resources required to transmit data via SMS. To ensure data redundancy, the system was equipped with flash memory where the data were stored for backup/later recovery. Overall, it appears that there is value in performing transmissions only when an event occurs (discussed later), yet one must enable robust checking/handshaking to ensure delivery.

### 4.2. Event Detection

Table 2 presents summary statistics for the landfill and home energy event data sets. It should be noted that the minimum sensed value and the minimum event duration are dependent on the set threshold level for event identification and on the sampling frequency.

With respect to the CO${}_{2}$ event statistics, it can be seen that the total number of events is high at 70 (should ideally be 0 for full compliance) with an average of ca. 6 events per month. Furthermore, the average CO${}_{2}$ concentration during the detected ‘events’ was 9%, and in some cases, events lasted longer than 8 days.

With respect to the home energy usage, the RMS current measurement results show the frequency of shower events was, on average, 1.5 per day, with an average power draw of ca. 32 A and an average duration of ca. 7 min. This is important for calculating how much the shower alone is costing the household and if it is a high proportion of the electrical bill (in one case, the shower was active for 33 min!—as seen in Figure 4).

### 4.3. Selectivity and Accuracy

Event classification in the Environmental dataset followed a relatively straightforward path guided by legal emission limits. In contrast, the energy monitoring task proved to be more complex, necessitating the use of a differential dataset. The differential approach underscores the significance of algorithms in conferring “selectivity” to decision-making processes based on signals from physical transducers. In essence, these algorithms serve a role akin to binding sites in chemical sensing. Their function is to recognize and report features in the electronic signal that align with predefined “event” patterns.

Similar to binding sites, algorithmic selectivity is not always absolute. A decision must be made regarding whether the event originates from the primary target signal or an interfering source. In both cases, practical compromises are necessary to address analytical challenges. Perfectly selective binding sites that only report the binding of the primary molecule under all conditions are a rarity, leading to inevitable occurrences of false positives and false negatives. Likewise, in appliance detection, complex algorithms may achieve near-perfect classification accuracy, but their implementation becomes impractical in distributed sensor networks, where computational resources are limited.

To put it differently, a certain degree of uncertainty is inherent in all elements of sensor networks. Our experience in both chemical sensing and data analytics has revealed similarities in the functioning of the two networks presented in this paper. Environmental IR sensors may encounter interferences from molecules with similar binding characteristics, just as appliance recognition may yield false positives when multiple appliances drawing power simultaneously create overlaps. The practical objective is to deliver data of sufficient quality to meet the application’s requirements. In smart metering, this means initially detecting activity versus inactivity and subsequently identifying appliances with acceptable accuracy. Combining relatively simple algorithms can achieve this goal, at least for power-hungry devices that generate distinct signature profiles in the root mean square (RMS) current when operated.

Furthermore, the integration of AI and ML for event detection introduces a compelling challenge. Overfitting, a common concern, can significantly impact the effectiveness of event detection models. Overfitting occurs when a model becomes excessively specialized in performing well on the training data but struggles to generalize its performance to new, unseen data [70,71]. This challenge is particularly relevant when dealing with large datasets, as previously mentioned. In the context of IoT event detection, striking a balance between avoiding overfitting and achieving generalization is essential for a robust and effective event detection system. The challenge lies in ensuring that the model performs well on the training data without becoming overly specialized and, as a result, less adaptable to different environments or domains.

Although the use of relatively simple algorithms for data processing on microcontrollers may exhibit limitations, they can demonstrate substantial value in green IoT event detection. Similar to the imperfections encountered in chemical binding and AI/ML approaches, some degree of uncertainty may persist in the data produced. However, it is crucial to recognize the potential value that sensor node-based data processing can bring, especially in the context of reducing overall carbon emissions in terms of data processing. It must be noted that the detection of the longest shower (33.17 min, Table 2) corresponds with the third shower duration in the training set in Figure 4, which supports the accuracy of our approach.

The primary events of interest, such as detecting primary appliances for carbon emissions in the electricity and heating sector or tracking environmental chemical sensing related to waste disposal, hold significant implications for addressing the critical challenge of climate change. Therefore, embracing the possible imperfections and exploring the possibilities offered by these low-power devices is a worthwhile pursuit with promising environmental and societal benefits.

#### 4.3.1. Accuracy of the Event Classification Algorithm

The accuracy of our event classification algorithm is a critical aspect of our study. We have made the algorithm’s source code, written in the C programming language, available in Table A1 in Section A.1 for comprehensive transparency. The algorithm operates in real time, processing data on the fly without the need for extensive memory storage, therefore minimizing memory requirements.

To assess the accuracy, we performed a manual comparison with the Environmental CO${}_{2}$ dataset. Although the Energy-Monitoring dataset contains a higher sampling frequency (1.5 min), the Environmental dataset, with a sampling period of 6 h, was chosen for this specific analysis due to the manageable number of data points for auditing. The manual processing of the Environmental data, when compared with the outcomes of our binary classification algorithm, demonstrated complete agreement. This meticulous verification process attests to the algorithm’s accuracy in identifying and classifying events.

#### 4.3.2. Advantages of the Event Classification Algorithm

Our event classification algorithm exhibits notable advantages. Operating autonomously, the algorithm processes data in real time, eliminating the need for extensive memory storage during its operation and contributing to reduced memory requirements. The device’s on-the-fly processing further minimizes its memory footprint, aligning with resource-efficient practices and memory constraints on microcontrollers. With a relatively high sampling frequency of 6 h for the Environmental dataset, our device offers a significant monitoring advantage over traditional manual methods, which often operate on a much lower frequency (e.g., 4 times per year). This autonomous capability allows for continuous monitoring, showcasing its practicality in environmental applications. Our validation of our algorithm on the Environmental dataset resulted in a 100% agreement in event statistics. This reliability underscores the effectiveness of our algorithm in field measurements. Importantly, our algorithm’s efficiency reduces reliance on manual data collection and the deployment of highly advanced machine-learning systems. By utilizing the processing capabilities of the microcontroller, our device provides critical information with high accuracy, contributing to a reduction in carbon emissions associated with complex computational processes.

### 4.4. Power Impact

One of the critical factors in the design of deployable sensor networks is power consumption. Transmitting data consumes a significant amount of energy, especially when using battery-powered devices. To address this issue, we explore a scenario where the sensor network minimizes data transmission when no events occur, as indicated by the deployment statistics presented in Table 1 and Table 2.

#### 4.4.1. Deployments

First, let us examine the environmental CO${}_{2}$ data, where a total of 70 events were detected out of 1121 transmitted measurements over the year. If the system were configured to transmit data only when an event occurs, an impressive 93.8% of measurements could be spared from transmission, resulting in substantial power savings. Alternatively, even if data were transmitted at the beginning and end of an event, it would still mean that 87.5% of the transmitted data were redundant, representing a significant energy waste. This aspect becomes particularly critical for battery-powered devices, especially when utilizing power-hungry communication methods like GSM.

In contrast, the energy data presents an even more pronounced opportunity for power conservation. A staggering 99.8% of the transmitted data are considered redundant if data transmission occurs only once per event. Even with data transmitted at the start and end of an event, 99.7% of the transmitted data remains unnecessary. This high percentage of redundant data underscores the significant energy waste in the context of energy monitoring. However, it is worth noting that, unlike environmental data, the tracking of significant appliance usage may not require real-time alerts. Thus, in the context of energy monitoring, a single transmission per event could be a practical and energy-efficient approach, balancing the need for data with power conservation.

In the current paradigm of sensor networks, the prevalent approach often involves collecting raw data and transmitting it to the cloud for later processing using AI/ML techniques. However, this approach can result in unnecessary processing and energy consumption. By adopting a more efficient strategy, as demonstrated in this study with event detection on the microcontroller itself, significant benefits can be achieved. Not only does this approach lead to reduced data transmission, prolonging the operational lifespan of the sensor, but it also contributes to the reduction of carbon emissions. The need for continuous server farms can be minimized, aligning with the broader goal of addressing climate change by making resource allocation more sustainable and environmentally friendly.

#### 4.4.2. Power Savings

To assess the power savings achieved through our event detection method, we conducted a detailed analysis of the power usage for the two deployments. The results presented in Table 3 highlight the significant reduction in energy consumption when the platform is not required to transmit until the end of an event.

In the context of environmental CO${}_{2}$ monitoring, the average current during transmission (TX) was measured at 37 mA, with an average TX time of 22.9 s. The energy consumption per TX was calculated as 10.1 J, resulting in an overall energy consumption of 11.4 kJ for all TX events. However, by leveraging our event detection method, the energy expended for events (TX/RX) was substantially reduced to 710.2 J, leading to an impressive energy savings of 93.8%.

Similarly, in the case of energy monitoring, where the platform transmits only to report an event, the energy savings were even more remarkable. With an average TX current of 35.5 mA and a minimal TX time of 0.1 s, the energy consumption per TX was merely 19.2 mJ. The overall energy consumption for all TX events was 6.6 kJ, but with the event detection method, the energy expended for events (TX/RX) reduced significantly to 10.3 J, yielding an extraordinary energy savings of 99.8%.

These findings underscore the effectiveness of our event detection method in minimizing energy consumption, making it a crucial component for sustainable and energy-efficient sensor network operations in addition to further reducing the carbon footprint. By extending this to a WSN of thousands of sensory devices, one can understand the significance of adopting a more power-efficient strategy.

#### 4.4.3. Platform Power Comparison

A traditional AI/ML system typically refers to a setup where machine-learning tasks, such as training and inference, are performed on powerful computational platforms, often involving high-performance servers or cloud-based infrastructure. These systems may include Graphics Processing Units (GPUs) or specialized hardware like Tensor Processing Units (TPUs) to accelerate the computational demands of machine-learning algorithms. In the context of this work centralized around carbon emissions and generation, it becomes important to compare the various platforms in terms of power use. Table 4 presents the power draw of the platforms used in this study and alike with other platforms typically used for AI/ML with conversion to carbon emissions using the USA current definition of 367 gCO${}_{2}$e [72]. We note that the first two platforms in the table were used in this study: MSP430 (Case Study I) and EMP250 (Case Study II). Although server farms clearly use significantly more power than those listed, they were omitted for closer comparisons, i.e., it was more comparable to list microcontrollers with edge devices and desktop equivalents. A stark contrast exists between the power draw of microcontrollers (top 3) and CPU/GPU-based platforms (bottom 6), which draws approximately ×20,000 the amount of power as microcontrollers, resulting in the same proportion of carbon emissions.

### 4.5. Energy Harvesting

In the context of addressing carbon emissions, it is essential to consider the sustainability of sensor nodes, particularly their ability to operate using renewable energy sources, such as solar cells. Sensor networks are uniquely positioned to operate for extended periods, thanks to the low-power characteristics of microcontrollers, which complement the energy output of available energy harvesting methods. This stands in stark contrast to the resource-intensive operation of server farms. Although significant advancements are being made in harnessing renewable energy sources for data centers [79], the demand on server farms remains such that achieving zero carbon emissions is still a challenge.

Alternative approaches to sustainable sensing systems involve strategies aimed at significantly reducing the per-sample power cost. For instance, opting for reagent chemistry with LEDs for gas detection [63,80], as opposed to IR-based methods, can lead to substantial power savings. We note that the use of LEDs as light detectors has demonstrated significantly less power consumption than the traditional photodiode counterpart [81,82,83]. For instance, previous work in turbidity sensing [84] performed a direct comparison between photodiodes (447 mW) and LED (446 nW) power draw and found two orders of magnitude difference, i.e., 1.002 ×$10^{-6}$ less power draw. Such advances, coupled with renewable energy strategies, could prove invaluable for battery-powered devices.

In scenarios where water-based sensors are employed [85,86], especially in environments with limited access to renewable energy sources such as solar or wind, an event-driven approach proves advantageous. The attenuation of RF signals in water can hinder data transmission, making it essential to minimize data transmission and adopt an event-driven model. Additionally, alternative communication methods like sonar, while effective, can be power-intensive, and their usage should be optimized to maximize the longevity of deployments.

By integrating energy harvesting and optimizing power usage in sensor networks, we can further reduce the carbon footprint associated with continuous data transmission and processing, contributing to a more sustainable and environmentally responsible approach to sensor network design.

### 4.6. Resource Placement

Historically, the development of sensor systems necessitated a myriad of specialized resources, often involving the expertise of electrical engineers, intricate PCB design and assembly, and the creation of custom firmware. These requirements made entry into the field of sensor networks daunting and limited to those with extensive technical knowledge.

However, a transformative shift occurred with the advent of single-board microcontrollers. These compact, versatile devices ushered in a new era of sensor system development, democratizing the process and making it accessible to a wider range of enthusiasts. A pivotal aspect of this transformation was the emergence of a vibrant community of users and the availability of libraries that offered pre-built code for common functions. As a result, the barriers to entry significantly lowered, enabling a more diverse group of individuals to engage in sensor network projects.

In recent times, there has been a prevailing trend toward harnessing the power of artificial intelligence (AI) and machine learning (ML) for data processing within sensor networks. The appeal of these technologies lies in their ability to perform complex tasks, offering high-level programming and classification solutions. AI and ML tools continue to evolve, providing increasingly sophisticated capabilities for handling data and making sense of intricate patterns.

Although the integration of AI and ML for data processing presents remarkable advantages, it is not without drawbacks. One of the significant downsides is the increased power consumption associated with these resource-intensive systems. Consequently, the carbon footprint of sensor networks can expand, which may counteract the overarching goal of reducing carbon emissions.

In this study, we have demonstrated the noteworthy achievements that can be unlocked through the utilization of microcontrollers as primary resources for gathering and processing data in sensor networks. Our research has highlighted the potential of these low-power yet efficient devices to extract valuable insights from data, particularly in contexts where the primary focus is the identification and management of high-power-consuming appliances—a critical aspect of reducing carbon emissions.

To achieve the best possible reduction in carbon emissions, it is imperative to optimize the placement of resources within sensor networks. This entails careful consideration of when and where to employ resource-intensive AI/ML systems and when to leverage the efficiency of microcontrollers. Software tools like openLCA offer a means to assess the environmental impacts associated with every stage of the life cycle of WSN systems, from sensor deployment to data processing on servers. By optimizing resource placement, we can strike a balance between data processing efficiency and environmental sustainability, aligning with the overarching goal of addressing climate change.

#### Applications and Advantages of Microcontroller Methods

The applications of MCU methods in sensor networks for carbon emissions monitoring are diverse and offer distinct advantages. In the context of this research, the implementation of MCU-based real-time event detection has been successfully demonstrated in two specific applications: environmental CO${}_{2}$ monitoring and home energy usage tracking.

In environmental CO${}_{2}$ monitoring, the MCU method excels in providing a low-power solution for gathering and processing data from sensors. The MCU’s ability to perform event detection locally on the sensor node minimizes the need for continuous data transmission, resulting in significant energy savings. This approach is particularly advantageous for remote or inaccessible deployment locations, where frequent data transmission may not be feasible and/or transmission costs are a bottleneck. The MCU’s efficiency (coupled with event-driven data transmission) ensures that only relevant information is transmitted, which reduces the overall power consumption of the sensor node.

For home energy usage tracking, the MCU method proves instrumental in detecting and classifying events related to high-power-consuming appliances. The efficiency of MCU-based algorithms allows for real-time processing of data, enabling the identification of specific appliance usage patterns. This capability is crucial for assessing the contribution of individual appliances to the overall carbon footprint of a household. The MCU’s low-power characteristics make it a suitable choice for deployment in smart homes, where continuous monitoring is essential for understanding energy consumption patterns.

The advantages of MCU methods extend beyond specific applications to address overarching challenges in sensor network design. The inherent low-power characteristics of microcontrollers contribute to prolonged sensor node operational lifespans, reducing the frequency of battery replacements and associated environmental impact. Additionally, the accessibility and affordability of microcontrollers have democratized sensor system development, making it feasible for a broader range of individuals and communities to engage in carbon emissions monitoring initiatives.

In summary, the applications of microcontroller methods in carbon emissions monitoring encompass diverse scenarios. In this work, we demonstrated this from environmental sensing to household energy tracking. The inherent advantages of low-power real-time processing contribute to energy efficiency, sustainability, and accessibility in sensor network design. These attributes position microcontrollers as key components in addressing the challenges of climate change through effective resource placement and optimized data processing strategies.

### 4.7. Current and Future Challenges

As outlined in the Introduction, MCU-based data processing comes with various challenges, such as hardware heterogeneity, diverse MCU architectures, resource limitations, constrained memory management, and the need for seamless software interoperability across devices. These challenges can be addressed through the development of standardized programming libraries and compatibility with device-specific code compilation. This has been demonstrated through the developed code in Table A1 in Section A.1 for event detection for two diverse applications. It is noted that this was 98 lines of C code, while the same result can be achieved through a few lines in Python due to the relatively plentiful availability of memory and imported libraries.

Network connectivity remains a significant challenge for MCU-based solutions. Emphasis should be placed on standard protocols, interoperability, and security to overcome this challenge. The LwM2M protocol is discussed as a potential solution for addressing the challenges related to network connectivity and interoperability. Additionally, the first ACM/IEEE TinyML Design Contest (TDC’22) focuses on developing AI/ML-based real-time detection algorithms for life-threatening ventricular arrhythmia on low-power microcontrollers used in Implantable Cardioverter-Defibrillators (ICDs) [87].

Another concern is security and data trustworthiness issues for IoT systems [88,89]. Critical assurances must be put in place for the environmental data, as landfill sites can be financially penalized for breaching legal limits and be appealed in court systems. Moreover, considering that many are privately owned sites, there is a vested interest in favoring sub-threshold values by the owners and/or higher for competitors. For the household data, access by unauthorized persons could reveal when the household is empty as seen in Figure 5 and therefore be vulnerable to burglary [90]. Parikh et al. [91] outlined opportunities and challenges related to wireless systems, particularly for smart grid applications, which is particularly relevant for our household monitoring system. In addition, a game-theoretic approach has been proposed by Abdalzaher et al. [92] to enhance security and data trustworthiness in IoT applications. This model focuses on clustered wireless sensor networks (WSNs) in IoT, addressing challenges such as data trustworthiness (DT) and power management. The repeated game model presented in the paper aims to enhance security against selective forwarding attacks, detect hardware failures in cluster members, and conserve power consumption due to packet retransmission.

Another challenge is incremental on-device learning, which allows both inference and training of models directly on MCU devices. A toolbox called TyBox is introduced to address this challenge and provide automatic design and code generation for incremental on-device classification models [93].

## 5. Conclusions

In conclusion, this research underscores the transformative role of low-power microcontrollers in shaping sustainable and energy-efficient sensor networks for carbon emissions monitoring. By showcasing the successful implementation of MCU-based real-time event detection in diverse applications, the study establishes the feasibility of minimizing power consumption without compromising the quality of environmental data. The power savings achieved through event-driven data transmission, coupled with the exploration of energy harvesting strategies, positions MCU-powered sensor nodes as key components in reducing the carbon footprint of sensor networks. The work not only addresses the challenges of resource allocation in sensor systems but also emphasizes the need for a balanced approach, judiciously integrating AI/ML for complex tasks while leveraging the inherent efficiency of microcontrollers. Ultimately, this research contributes to a more nuanced understanding of resource placement in sensor networks, offering a practical and sustainable framework for addressing the critical issue of climate change.

## Figures and Tables

**Figure 1 sensors-24-00162-f001:**
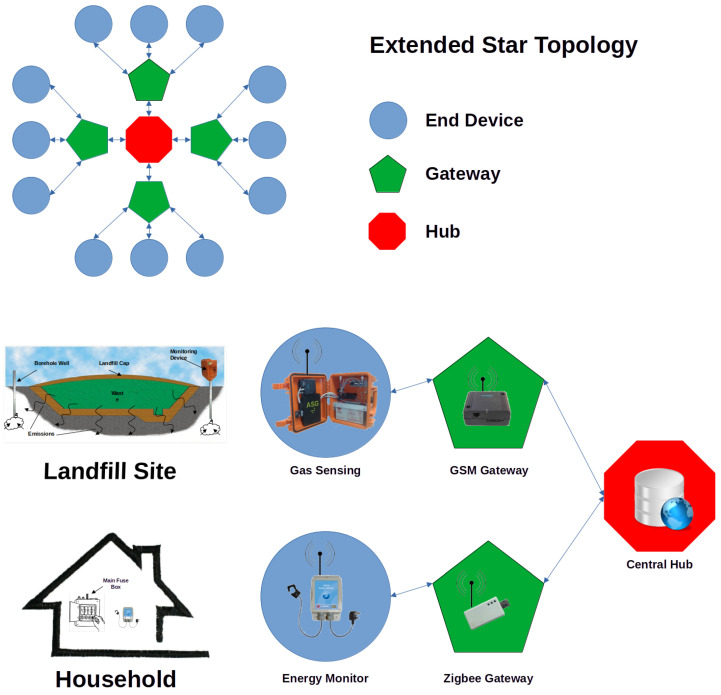
Sensing architecture showing the stages involved in harvesting data from the environmental CO_2_ sensor (**top**) and the home energy sensor (**bottom**). In each case, data were transferred wirelessly to a cloud-based central database repository via the Internet.

**Figure 2 sensors-24-00162-f002:**
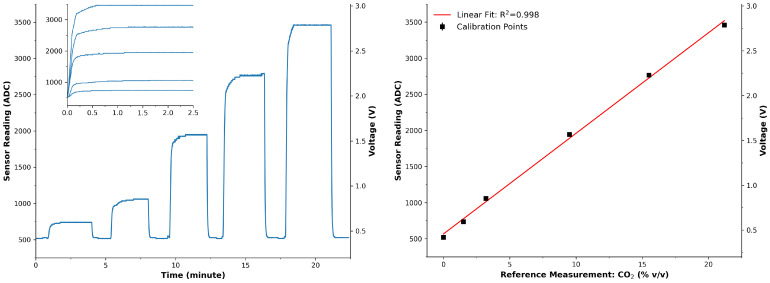
(**Left**): Response of the CO${}_{2}$ sensor during the calibration routine. Inset—extracted sections when the sensor was exposed to changing CO${}_{2}$ concentrations. (**Right**): Calibration model of the CO${}_{2}$ sensor. Black squares represent the average of the steady-state responses, error bars represent the standard deviation, and the red line is an excellent model applied to the data points (R${}^{2}$ = 0.998, n = 6); y = 139.275x + 567.197. Notably, there is a mismatch between the actual sensor responses (**left**) and the calibration model (**Right**). The steady-state response of the sensor response ((**left**) is calculated to represent the CO${}_{2}$ concentration and used to establish the conversion model (**Right**) from the microcontroller’s ADC value and the actual CO${}_{2}$ concentration measured using the reference instrument (GA2000 Plus).

**Figure 3 sensors-24-00162-f003:**
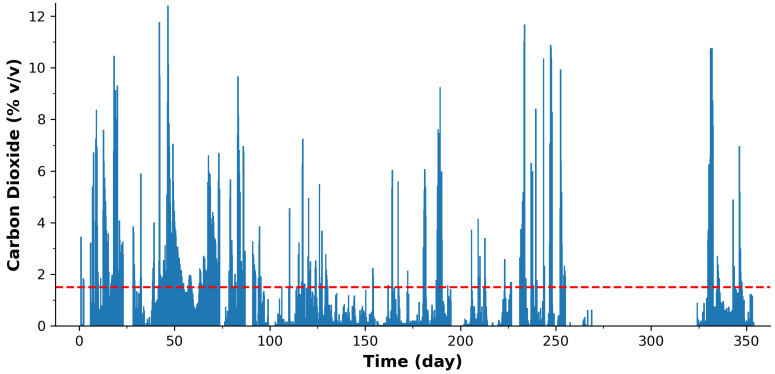
Measured CO${}_{2}$ concentration data for the 1-year (365-day) deployment. The Red dashed line represents the legal emission threshold.

**Figure 4 sensors-24-00162-f004:**
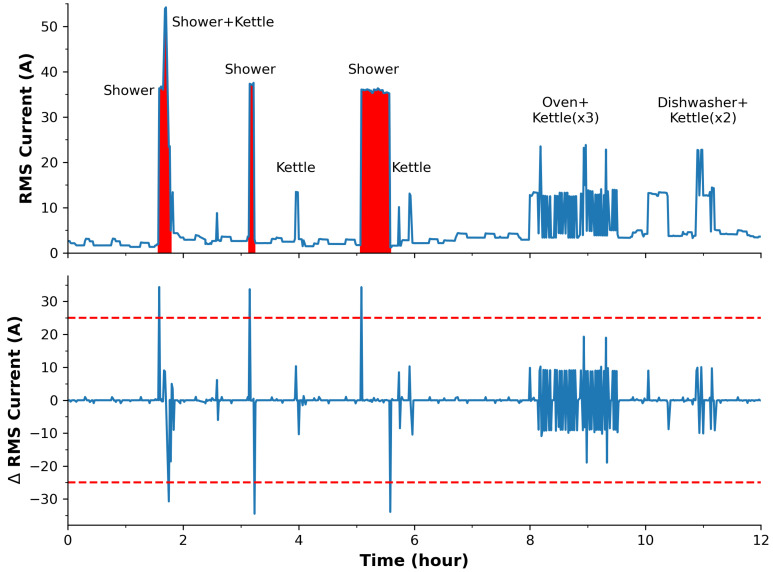
A 12-h appliance training set showing the global RMS current usage of the household and labels showing when each significant appliance was powered on. Labels X + Y indicate times when two appliances were powered on at the same time. **Top** plot shows the RMS current signal with red shaded regions when the shower was on. **Bottom** plot represents the differential data. The dashed red line shows the chosen threshold for detection of an event (shower switched on: 25 A, shower switched off −25 A). The background appliances, such as refrigerators and freezers, appear as smaller features on the baseline.

**Figure 5 sensors-24-00162-f005:**
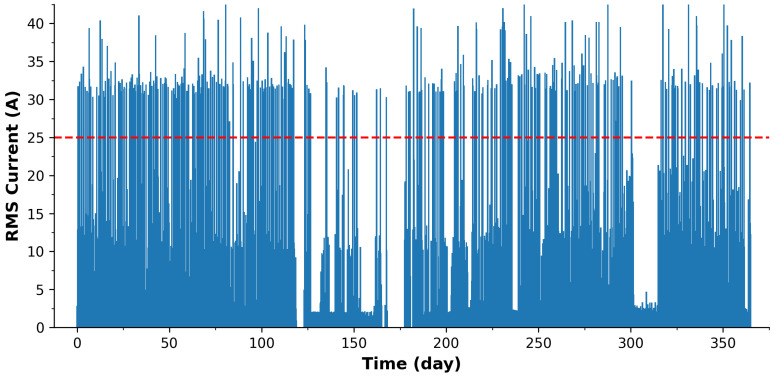
Plot showing the home energy consumption (RMS Current) for a 1-year deployment period. The red dashed line represents the threshold of 25 A. The spikes appearing above the threshold are likely representing the electrical shower in use.

**Table 1 sensors-24-00162-t001:** Statistical summary of events generated during both deployments.

Description	Environmental CO${}_{2}$	Energy Monitoring
Threshold	1.5%	25 A
Number of Measurements	1121	344,922
Summation	1591.59	1,168,664.6
Max Value	12.41%	48.64 A
Min Value	0.0%	0.69 A
Average	1.41%	3.39 A
% Measurements > Threshold	28.46	1.16

**Table 2 sensors-24-00162-t002:** Event statistical summary generated for both deployments.

Event Description	Environmental CO${}_{2}$	Energy Monitoring
Total Number of Events	70	539
Sampling Period	6 h	1.5 min
Avg. Events Per Day	0.192	1.47
Avg. Events Per Week	1.34	10.37
Avg. Events Per Month	5.75	44.92
Max Event Value	12.41%	48.64 A
Min Event Value	1.50%	28.03 A
Avg. Event Value	9.08%	32.90 A
Max Duration	8.25 days	33.17 min
Avg. Duration	1.05 days	7.25 min

**Table 3 sensors-24-00162-t003:** Power Use and Savings.

Description	Environmental CO${}_{2}$	Energy Monitoring
Avg. TX Current (mA)	36.992	35.5
Avg. RX Current (mA)	-	35.5
Avg. TX Time (s)	22.856	0.1
Avg. RX Time (s)	-	0.05
Energy per TX/RX (J)	10.146	0.0192
Energy for All TX/RX (J)	11,373.666	6622.502
Energy for Events TX/RX (J)	710.22	10.3488
Energy Savings (%)	93.756	99.844

**Table 4 sensors-24-00162-t004:** Platform comparison in terms of power requirements arranged from least (top) to most (bottom) power consumption.

Platform	Type	Power Draw (W)	gCO${}_{2}$e (US)/h	Reference
EM250	Microcontroller	$5.4\times 10^{-6}$	$1.98\times 10^{-6}$	[73]
MSP430 F449	Microcontroller	$616\times 10^{-6}$	$2.26\times 10^{-4}$	[74]
ATmega328P	Microcontroller	$750\times 10^{-6}$	$2.75\times 10^{-4}$	[75]
Raspberry Pi-Pico	Microcontroller	$6.5\times 10^{-3}$	$2.4\times 10^{-3}$	[76]
Raspberry Pi-4B	Microprocessor	2.56–7.30	0.94–2.68	[77]
NVIDIA Jetson Nano	Edge GPU	5–10	1.84–3.67	[77]
NVIDIA Jetson TX2	Edge GPU	7.5–15	2.75–5.51	[77]
NVIDIA Xavier	Edge GPU	8–15	2.94–5.51	[77]
Intel Core i7 8700	Desktop CPU	52–59	19.08–21.65	[78]
Nvidia RTX 2080Ti	Desktop GPU	102–157	37.43–57.62	[78]

## Data Availability

Restrictions apply to the availability of these data. Data were obtained from a private landfill site and can be requested from the EPA in the first instance.

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
