# Peer review of "Green IoT Event Detection for Carbon-Emission Monitoring in Sensor Networks"

_sensors, 2023, doi:10.3390/s24010162_

Round 1

Reviewer 1 Report

Comments and Suggestions for Authors

1, Please explain the accuracy of the event classification algorithm mentioned in the manuscript and its advantages.

2, Please provide a detailed explanation of the specific method for automatic analysis using microcontrollers mentioned in section 3.1.2 of the manuscript, as well as how the microprocessors process the data. It is recommended to provide a detailed explanation in the manuscript.

3, The manuscript mentions the potential of low-power microcontrollers as the main resource for data collection and processing, providing a sustainable and environmentally friendly method compared to traditional artificial intelligence and machine learning systems. Is there any data to support this claim? Please provide relevant data evidence.

4, Please explain the basis for the event detection method, the threshold level set for event recognition, and the setting of the sampling frequency.

5, Please provide a more accurate description of Figure 2, pointing out the mismatch between the model on the right side of the figure and the data on the left side. Additionally, please add the missing description content for Figure 5.

Reviewer 2 Report

Comments and Suggestions for Authors

1. Clarity of Methodology:

Clarify the methodology used for implementing the efficient use of microcontrollers in data gathering and processing. Provide more details on how these microcontrollers are integrated into the sensor networks. A step-by-step explanation of the processes and protocols involved would enhance the reader's understanding of the proposed solution.

2.Quantification of Power Savings:

Quantify the potential power savings achieved by using microcontrollers compared to traditional AI/ML systems. Providing specific data on the reduction in energy consumption and carbon footprint will strengthen the paper's argument and help readers grasp the practical implications of the proposed approach.

3.Real-world Examples and Case Studies:

Include real-world examples or case studies demonstrating the successful implementation of microcontrollers in sensor networks for carbon emissions monitoring. Concrete examples will not only support your findings but also make the paper more applicable and relatable to practitioners in the field.

4.Discussion on Potential Challenges:

Address potential challenges or limitations associated with the use of microcontrollers in sensor networks. Discussing issues such as scalability, adaptability to diverse environments, or any trade-offs made during the implementation will add depth to the paper and contribute to a more comprehensive understanding of the proposed solution.

5.Further Insights on AI/ML Tools:

Elaborate further on the judicious use of AI/ML tools in conjunction with microcontrollers. Provide insights into specific scenarios where AI/ML can complement the efficiency of microcontrollers without compromising environmental sustainability. This will strengthen the argument for a balanced approach in resource allocation.

6.Refinement of Abstract and Conclusion:

Refine the abstract and conclusion to succinctly capture the core contributions and findings of the research. Ensure that the abstract serves as an accurate summary of the paper, highlighting the unique perspective on resource allocation and the potential for environmental impact. The conclusion should reiterate the key takeaways and emphasize the significance of the proposed approach in the broader context of carbon emissions monitoring.

Reviewer 3 Report

Comments and Suggestions for Authors

No comments

Author Response

We wish to thank the reviewer for their time and effort in reviewing this manuscript.

Reviewer 4 Report

Comments and Suggestions for Authors

The manuscript has many flaws that should be carefully considered as follows:

The authors mentioned, "However, the prevailing trend towards the adoption of resource-intensive Artificial Intelligence (AI) and Machine Learning (ML) systems for data processing can paradoxically increase carbon emissions through energy-intensive server farms". However, the work done in the literature is against such a statement.

Afterward, they mentioned, "The findings also emphasise the need for optimising resource placement and employing AI/ML tools judiciously to strike a balance between data processing efficiency and environmental sustainability." It is a bit confusing.

The quality of Fig. 1 is poor. It is strongly recommended to redraw it.

The contribution is not clear?

Is it related to the use of microprocessors, AI, ML, or what?

What is the type of the used node in the present study?

What is the type of topology adopted?

What about the fading problems affecting data transmission?

The authors should show the effect of their model on the data retransmission.

The authors should show which ML model is adopted.

The authors should do more surveys on the related works. The following papers can assist the authors in the raised crucial comments: https://doi.org/10.1109/JIOT.2020.2996671; https://doi.org/10.1049/iet-com.2018.6272; https://doi.org/10.3390/s20216225

More results are strictly desired.

Comments on the Quality of English Language

Moderate editing of the English structure is required.

Round 2

Reviewer 1 Report

Comments and Suggestions for Authors Your answer is very detailed, but I still have doubts about some of the questions. Question1: Regarding the section on resource placement in the manuscript, please provide a detailed summary of the specific applications and advantages of the microcontroller methods in this area from the perspective of a peer reviewer. Question2: The manuscript does not provide a comprehensive introduction to microcontrollers. As a peer reviewer, what is your opinion on whether the hardware design aspect of microcontrollers should be included if the aim is to introduce the feasibility of microcontrollers?

Reviewer 4 Report

Comments and Suggestions for Authors

I have no further comments.

Comments on the Quality of English Language

 Moderate editing of the English language is required and can be done in proofreading.
